

# Tilapia lake virus causes mitochondrial damage: a proposed mechanism that leads to extensive death in fish cells

Promporn Raksaseri[1], Tuchakorn Lertwanakarn[2], Puntanat Tattiyapong[3], Anusak Kijtawornrat[4], Wuthichai Klomkleaw[1] and Win Surachetpong[3]

[1] Department of Anatomy, Faculty of Veterinary Science, Chulalongkorn University, Bangkok, Thailand
[2] Department of Physiology, Faculty of Veterinary Medicine, Kasetsart University, Bangkok, Thailand
[3] Department of Veterinary Microbiology and Immunology, Faculty of Veterinary Medicine, Kasetsart University, Bangkok, Thailand
[4] Department of Physiology, Faculty of Veterinary Science, Chulalongkorn University, Bangkok, Thailand

## ABSTRACT

**Background.** Tilapia lake virus (TiLV), also known as *Tilapinevirus tilapiae*, poses a significant threat to tilapia aquaculture, causing extensive mortality and economic losses. Understanding the mechanisms and pathogenesis of TiLV is crucial to mitigate its impact on this valuable fish species.

**Methodology.** In this study, we utilized transmission electron microscopy to investigate the ultrastructural changes in E-11 cells following TiLV infection. We also examined the presence of TiLV particles within the cells. Cellular viability and mitochondrial functions were assessed using MTT and ATP measurement assays and mitochondrial probes including JC-1 staining and MitoTracker™ Red.

**Results.** Our findings provide novel evidence demonstrating that TiLV causes cytotoxicity through the destruction of mitochondria. Transmission electron micrographs showed that TiLV particles were present in the cytoplasm of E-11 cells as early as 1 h after infection. Progressive swelling of mitochondria and ultrastructural damage to the cells were observed at 1, 3 and 6 days post-infection. Furthermore, losses of mitochondrial mass and membrane potential (MMP) were detected at 1 day after TiLV inoculation, as determined by mitochondrial probes. The results of the MTT assay also supported the hypothesis that the cell deaths in E-11 cells during TiLV infection may be caused by the disruption of mitochondrial structure and function.

**Conclusions.** Our study reveals the significant role of mitochondrial disruption in contributing to cellular death during the early stages of TiLV infection. These findings advance the understanding of TiLV pathogenesis and further enhance our knowledge of viral diseases in fish.

Corresponding author
Win Surachetpong, fvetwsp@ku.ac.th

# INTRODUCTION

Tilapia lake virus disease (TiLVD) is an emerging disease caused by tilapia lake virus (TiLV) that currently affects global tilapia aquaculture (*Eyngor et al., 2014*; *Surachetpong, Roy &*

*Nicholson, 2020*). TiLV was first described in tilapia from the Sea of Galilee, Israel in 2014, and concurrently, another virus causing high mortality and syncytial hepatitis of juvenile tilapia (SHT) was described in Ecuador (*Ferguson et al., 2014*). Subsequent studies have shown that SHT and TiLV share 98–100% genetic sequence identity (*Del-Pozo et al., 2017*), and is therefore the same virus causing disease in tilapia (*Bacharach et al., 2016*; *Ferguson et al., 2014*). TiLV, also known as *Tilapinevirus tilapiae*, is classified as a member of the family *Amnoonviridae* (*ICTV, 2022*; *Sunarto et al., 2022*). The virus has a spherical shape with a trilaminar capsid-like structure (*Del-Pozo et al., 2017*).

TiLV primarily infects tilapia and its hybrid species (*Surachetpong, Roy & Nicholson, 2020*; *Waiyamitra et al., 2021*); however, other cichlid fishes including Giant gourami (*Osphronemus goramy*) (*Jaemwimol et al., 2018*), ornamental African cichlids (*Aulonocara* sp.) (*Yamkasem et al., 2021*), and angel fish (*Pterophyllum scalare*) (*Paria et al., 2023*) have been found to be susceptible to TiLV infection. Clinical signs of TiLV infection include erratic swimming, skin hemorrhage, exophthalmos, abdominal swelling, anemia, and scale protrusion (*Eyngor et al., 2014*; *Ferguson et al., 2014*; *Tattiyapong, Dachavichitlead & Surachetpong, 2017*; *Turner et al., 2023*). Microscopic examination of infected tilapia has revealed inflammation and necrosis of various organs, including liver, spleen, head kidney, gills, and brain tissues (*Mugimba et al., 2018*; *Pierezan et al., 2020*; *Tattiyapong, Dachavichitlead & Surachetpong, 2017*). However, the mechanism behind cell death caused by TiLV infection is currently not fully understood.

Previous studies have demonstrated that TiLV can multiply and lead to cell death in various fish cell lines (*Eyngor et al., 2014*; *Lertwanakarn et al., 2021*; *Li et al., 2022a*; *Thangaraj et al., 2018*; *Wang et al., 2018*; *Yadav et al., 2021*). A recent study by *Lertwanakarn et al. (2021)* revealed that TiLV infection caused cytopathic effect (CPE) in E-11 cells within 3 days, while CPE formation in other cell lines may vary between 3 to 11 days post-infection (dpi) (*Li et al., 2022a*; *Thangaraj et al., 2018*; *Wang et al., 2018*; *Yadav et al., 2021*). Transmission electron microscopy (TEM) has revealed that TiLV particles are rounded in shape and approximately 60–110 nm in size (*Eyngor et al., 2014*; *Li et al., 2022a*; *Piewbang et al., 2022*; *Thangaraj et al., 2018*; *Wang et al., 2018*). These particles can be found in susceptible cells as early as 3 dpi (*Piewbang et al., 2022*). Additionally, ultrastructural changes, such as swollen Golgi apparatus and mitochondria, as well as dense chromatin nuclei have been observed in TiLV-infected cells (*Del-Pozo et al., 2017*; *Ferguson et al., 2014*). However, the dynamic studies of TiLV on mitochondrial structure and functions have never been investigated.

Mitochondria play a crucial role in energy metabolism and the oxidative stress response in cells. In fish, mitochondria are responsible for oxygen consumption, as evident from the presence of mitochondrial-rich cells in the gills of most teleosts (*Lin & Sung, 2003*). Furthermore, mitochondria are involved in apoptosis, a process of cell death. The cold stress response in Nile tilapia has been associated with reduced mitochondrial membrane potential (MMP) and ATP production, leading to cellular apoptosis in various organs (*Liu et al., 2022*). Additionally, pendimethalin toxicity has been shown to cause mitochondrial defects in tilapia due to excessive oxidative stress, leading to damage in the brain, spleen, and gills (*Nassar, Abdel-Halim & Abbassy, 2021*). Similar to TiLV, infection with other piscine

viruses such as the infectious spleen and kidney necrosis virus (ISKNV) in the grouper cell line GF-1 can also lead to a deterioration in mitochondrial membrane potential (MMP), increased oxidative stress, and cell apoptosis and tissue damage (*Chen et al., 2022*). Importantly, the disruption of mitochondrial function is one of the key mechanisms that can lead to cell apoptosis and organ failure during viral infection and chemical toxicity. Despite its widespread distribution, there is a limited understanding of the biology and host cell alteration caused by TiLV infection. However, understanding the mechanism by which TiLV causes cell damage and changes in cellular functions can provide insights into the pathogenesis of this important virus, particularly its impact on mitochondria. Furthermore, this knowledge can be used to develop strategies to prevent and manage TiLV infections in fish populations.

In this study, we investigated the pathogenesis of TiLV infection involving the mitochondrial disruption in fish cells using TEM, cellular viability assays, and mitochondrial probes. Our findings suggest that TiLV infection leads to mitochondrial damage, impairs MMP, and induces cytotoxicity.

## MATERIALS AND METHODS

### Propagation of TiLV

The TiLV strain VETKU-TV01, previously isolated from the brain of infected red tilapia (*Oreochromis sp.*) (*Tattiyapong, Dachavichitlead & Surachetpong, 2017*), was used in the study. E-11 cells, a clone of SSN-1 cells isolated from snakehead fish (*Ophicephalus striatus*) (*Iwamoto et al., 2000*) were obtained from the European Collection of Authenticated Cell Cultures (ECACC), England (catalog number 01110916). E-11 cells were cultured in Leibovitz's L-15 medium supplemented with 5% fetal bovine serum (Sigma, USA) and 2 mM L-glutamine at 25 °C in a $CO_2$-free environment until they reached 70–80% confluence. The culture medium was subsequently removed, and inoculated with a viral load of 0.1 MOI for 1 h at 25 °C. Following incubation, the virus suspension and culture medium were aspirated, and the cells were thoroughly rinsed. Cells were maintained in Leibovitz's L-15 supplemented with 2% fetal bovine serum (Sigma, USA) and 2 mM L-glutamine at 25 °C in a $CO_2$-free environment. Daily microscopic observations were conducted until 80% of the cells exhibited CPE. The protocol for handling the virus was approved by the Institutional Biosafety Committee (IBC), Faculty of Veterinary Medicine, Kasetsart University under the protocol number IBC-63-V02.

### Virus purification by glucose gradient centrifugation

Once 80–100% CPE formation was observed, infected E-11 cells were disrupted using three rounds of freeze-thaw cycles followed by centrifugation at 3,000× g for 10 min. The supernatant containing TiLV was collected and stored at −80 °C until further use. The supernatant was thawed and re-suspended in a 30% sucrose solution in 14 × 89 mm thin wall polypropylene centrifuge tubes (Beckman Coulter, USA). The suspension was then centrifuged at 40,000 rpm (10,000× g) for 1.5 h at 4 °C using an Optima L-90K Ultracentrifuge (Beckman Coulter, Brea, CA, USA). The pellet was collected and resuspended in one mL of TN buffer (0.1 M NaCl, 0.01 M Tris pH 7.4). The suspension

was overlaid on top of a glucose gradient solution consisting of two mL layers of 30%, 40%, and 50% (w/v) sucrose in TNE buffer (0.1 M NaCl, 0.01 M Tris pH 7.4, 3 mM EDTA) and subjected to centrifugation at 40,000 rpm for 1.5 h at 4 °C. Two mL of each fraction was collected and mixed with 10 mL of phosphate buffer saline (PBS) buffer. To remove excess sucrose solution, the suspension was ultracentrifuged at 40,000 rpm for 30 min at 4 °C. The supernatant was discarded, and 500 μL of PBS solution was added to each fraction, which was then stored at 4 °C until further analysis.

## Transmission electron microscopy (TEM) with positive staining

Uninfected and infected E-11 cells were collected at 0, 1, 3, and 6 dpi. Cells were trypsinized from the culture flasks and transferred to a 1.5 mL tube at room temperature (25 °C). The cell suspension was centrifuged at 2,000 rpm for 5 min at 4 °C (Centrifuge 5418R; Eppendorf, Hamburg, Germany). The supernatant was discarded, and the pellet was resuspended in 500 μL of 0.1 M PBS, followed by incubation in 2.5% glutaraldehyde in 0.1 M PBS at 4 °C overnight. The following day, the samples were thoroughly rinsed with 0.1 M PBS for 10 min, three times. The cell pellets were then incubated with 1% osmium tetroxide in $dH_2O$ for 1 h according to a previous protocol (*Barreto-Vieira & Barth, 2015*). The pellets were rinsed with $dH_2O$ for 10 min three times, dehydrated in acetone, and embedded in resin. Ultrathin sections were prepared by cutting samples at 90 nm thick using a glass knife. Samples were placed on a thin copper grid for 15 min and stained with 5% uranyl acetate for 15 min and lead citrate for 15 min. Samples were examined under a Hitachi HT7700 transmission electron microscope (Hitachi) at the Scientific Equipment and Research Division, Kasetsart University, Bangkok, Thailand. All micrographs were taken at 80 kV.

## Transmission electron microscopy (TEM) with negative staining

The purified TiLV was suspended in PBS (4 °C) and then transferred to Formvar® film-coated copper grids with 400 mesh sizes (Electron Microscopy Science, Hatfield, PA, USA) for 30 min. The grids were washed with $dH_2O$ before being stained with 40 μL of 2% uranyl acetate or 1% phosphotungstic acid for 1 min using filtered paper to remove the staining solution. The samples were dried for 7 days and then observed with the TEM operating at 80 kV.

## Cell viability assay

The cell viability was determined by 3-[4,5-dimethylthiazol-2-yl]-2,5-diphenyl tetrazolium bromide (MTT) assay. Briefly, E-11 cells were seeded in a 96-well plate at a density of $4 \times 10^4$ cells/mL per well and incubated overnight with L-15 medium supplemented with 5% FBS at 25 °C without $CO_2$. The cells were then infected with TiLV at 0.1 MOI for 1 h in a humidified incubator at 25 °C. Next, cells were incubated with MTT at a concentration of 5 mg/mL in L-15 media for 4 h in a 37 °C humidified incubator. The uninfected and infected E-11 cells were collected at 1 h (0 dpi), 1, 3, and 6 days after TiLV inoculation and used as control and experimental groups, respectively. The media containing MTT were then removed and replaced with 100% DMSO to solubilize formazan, and the absorbance of the solubilized formazan in each group was measured using a hybrid multi-mode

microplate reader (Synergy™ H1; Agilent, Santa Clara, CA, USA) at a wavelength of 590 nm.

## Measurement of ATP concentration

The ATP concentration of the cells was evaluated using the CellTiter-Glo® luminescent cell viability assay (Promega, Madison, WI, USA). E-11 cells were incubated at 25 °C, 100% $O_2$ overnight. The cells were then treated with TiLV at 0.1 MOI ($n = 3$) for one hour before being replaced with Leibovitz's medium containing 2 fetal bovine serum (2% FBS L-15). Cells treated with 2% FBS L-15 and blank were included as negative controls. The measurement of ATP concentration was performed on 0, 1, 3, and 6 dpi, by extrapolating from the ATP (Sigma, St Louis, MO, USA) standard curve (10–1,000 nM). Luminescence was detected using a luminometer (Synergy H1™, BioTek® Instruments, Inc., Winooski, VT, USA) for 1 s at 37 °C, and the relative light units were read using Gen5™ software (BioTek® Instruments, Inc., Winooski, VT, USA).

## Detection of red-to-green ratio in JC-1-stained E-11 cells

The mitochondrial function of E-11 was evaluated using the fluorochrome 5,5′,6,6′-tetrachloro-1,1′,3,3′ tetraethylbenzimidazolyl-carbocyanine iodide (JC-1) (Molecular probes Inc, USA). E-11 cells were plated in 24-well flat-bottom plates and allowed to reach 70–100% confluence. The cells were then infected with 0.1 MOI of TiLV for 1 h, followed by the replacement of 2% FBS L-15, and further incubation at 25 °C. Uninfected E-11 cells were used as a control. At 0, 1, 3, and 6 dpi, the cells were incubated with 5 µM of JC-1 for 30 min, washed with PBS twice, and visualized under an inverted fluorescence microscope (IX73, Olympus, Japan). Green and red images were captured and analyzed under the BW channel (bandpass 460–495 nm; barrier filter 510 nm; dichroic mirror 505 nm) and GW channel (bandpass 530–550 nm; barrier filter 575 nm; dichroic mirror 570 nm), respectively. All pictures were merged into a new image using cellSens dimension™ 2.3 software (Olympus, Tokyo, Japan). The intensity of green and red colors was randomly analyzed from three dispersed areas with more than 1,000,000 pixels, and the average red-to-green (R/G) ratios were calculated and compared between uninfected and TiLV-infected E-11 cells.

## Determination of mitochondrial mass in E-11 cells

The number of mitochondria was investigated using MitoTracker™ Red CMXRos staining (Invitrogen™, Eugene, OR, USA). The E-11 cell line was cultured in Leibovitz's L-15 media with 5% FBS at 25 °C without $CO_2$. After trypsinization and cell counting, the cells were seeded onto collagen-coated cover slips and allowed to incubate until they reached 80–90% confluence. Subsequently, the cells were inoculated with TiLV at 0.1 MOI and then stained with MitoTracker™ in a dark room. Following staining, the cells were fixed with methanol and permeabilized using Triton X-100 (Sigma-Aldrich, St Louis, MO, USA). To visualize the cell nuclei, the cells were further incubated with 4′,6-diamidino-2-phylindole (DAPI) at a concentration of 1:1,000. Representative images were acquired using confocal laser scanning microscope model FLUOVIEW FV3000 (Olympus, Tokyo, Japan) with specific

filters for MitoTracker™ Red (579 nm excitation/599 nm emission), and DAPI (358 nm excitation/461 nm emission).

## Statistical analysis

All data were statistically analyzed using GraphPad™ Prism software (San Diego, CA, USA). Results were shown as the mean ± standard error of the mean (S.E.M.). The data were tested for normal distribution using Kolgomorov-Smirnov test, and all data followed a Gaussian distribution. The cell viability, ATP measurement and JC-1 red-to-green ratios were compared between uninfected and TiLV-infected at 0, 1, 3, and 6 dpi using two-way ANOVA, followed by Tukey's as a post-hoc test. Statistical significance was considered at $p$-value less than 0.05.

# RESULTS

## TiLV caused morphological changes and cytopathic effects in E-11 cells

The infection of E-11 cells with TiLV resulted in significant morphological changes and cytopathic effects (CPE). Within 24 h of infection, the infected cells showed altered morphology, with CPE progression from 10% to 90% between 1 to 6 days post-infection (dpi). At 1 dpi, uninfected cells remained normal (as seen in Fig. 1A), while infected cells exhibited vacuolation and pyknotic nuclei (as seen in Fig. 1B). By 3 dpi (as seen in Fig. 1C) and 6 dpi (as seen in Fig. 1D), extensive vacuolation, shrinkage, and distinct CPE formation were observed, with only a limited number of viable cells remaining and discoloration of the culture media at 6 dpi. These findings demonstrate that TiLV infection leads to substantial changes in the morphology and viability of E-11 cells.

## Ultrastructural changes of E-11 cells during TiLV infection

The ultrastructure of uninfected and TiLV-infected E-11 cells was compared using TEM. Figures 2A and 2C depict the ultrastructure of uninfected cells, while Figs. 2B and 2D demonstrate the ultrastructure of TiLV-infected cells. Within 1 h post-infection (0 dpi), a viral particle was observed at the plasma membrane of the infected E-11 cells (Fig. 2B; inset). However, no significant changes in the cellular structure and organelles were noticed at this early time point. Both uninfected and TiLV-infected cells exhibited intact nuclear membranes, normal mitochondria (Figs. 2C and 2D), and typical rough endoplasmic reticulum (rER) at 0 dpi (Figs. 2A, 2B, and 2C).

At 1 dpi, the uninfected cell displayed both normal mitochondria and mitochondria with partial loss of cristae (Fig. 3A). In contrast, TiLV-infected cells exhibited initial changes, including swollen mitochondria, indistinct mitochondrial membrane structure, and cristae degeneration (Fig. 3B). At 3 dpi, cristae remained visible in the mitochondria of uninfected cells (Fig. 3C), while progressive mitochondrial degeneration was observed in TiLV-infected cells, characterized by extensive cristae loss. Furthermore, TiLV-infected cells displayed the formation of lamellar bodies and a large number of free TiLV particles (Fig. 3D). At 6 dpi, uninfected cells still maintained intact mitochondrial membranes and cristae (Fig. 3E). In contrast, TiLV-infected cells exhibited multiple cytoplasmic vacuolations. At this time

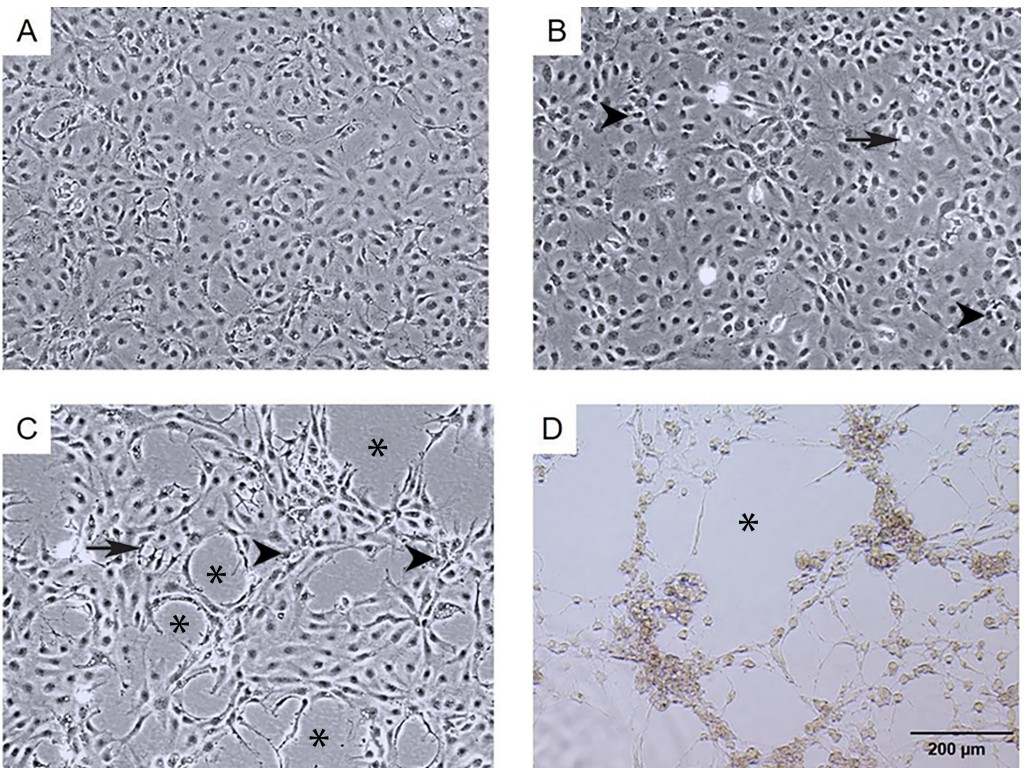

**Figure 1** **Morphological changes and cytopathic effects of TiLV infection in E-11 cells.** (A) Uninfected E-11 cells showing a normal appearance and confluence (B) TiLV-infected E-11 cells at 1 dpi displaying vacuolation (arrow) and pyknotic nuclei (arrowhead) (C) TiLV-infected E-11 cells at 3 dpi exhibiting extensive cell vacuolation, cell shrinkage, and cytopathic effects (CPE; asterisks). (D) TiLV-infected E-11 cells at 6 dpi showing complete cell detachment and CPE formation.

point, the mitochondria showed progressive degeneration, including swelling, major structural distortion, delamination of the inner and outer mitochondrial membranes, and complete loss of cristae. Intracytoplasmic TiLV particles were also prominently present (Fig. 3F). Additionally, the ultrastructure of isolated TiLV particles was examined in Fig. S1. The particles were found to have a round or oval shape, with diameters ranging from 50 to 120 nm. The particles exhibited a central electron-dense core surrounded by a capsid-like bilaminar structure. Notably, the spike protein was not observed on the surface of the TiLV particles.

## TiLV caused extensive mitochondrial damage and cell death

To evaluate the impact of TiLV infection on cellular viability and mitochondrial damage in E-11 cells, the MTT assay, ATP measurement, and JC-1 staining were employed (Fig. 4). The MTT assay revealed that TiLV infection resulted in significant cell death, with a progressive decline in the number of viable cells from $104.39 \pm 5.85\%$ at 0 dpi to $6.89 \pm 7.21\%$ at 6 dpi (Fig. 4A). Likewise, the amount of ATP concentration in E-11 cells following TiLV infection gradually reduced from $1.01 \pm 0.01\,\mu M$ at 0 dpi to $0.77 \pm 0.03\,\mu M$ at 6 dpi (Fig. 4B). Notably, the JC-1 staining demonstrates the alteration of

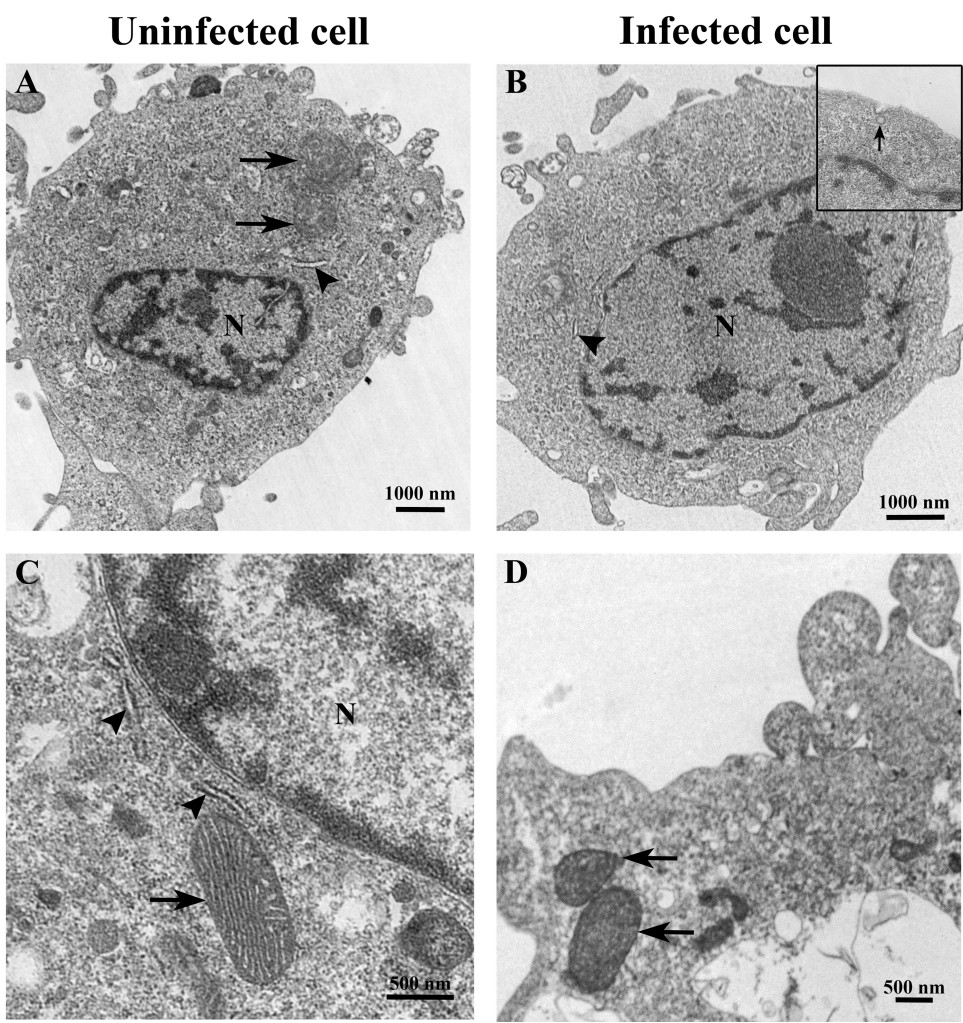

**Uninfected cell**       **Infected cell**

**Figure 2 Representative TEM micrographs of uninfected and TiLV-infected E-11 cells at 0 dpi (1 h post-TiLV inoculation).** (A) Uninfected E-11 cell with normal mitochondria (arrows), and nucleus (N). (B) TiLV-infected E-11 cell with normal nucleus (N), rough endoplasmic reticulum (rER: arrowhead), and presence of intracytoplasmic TiLV particle close to the plasma membrane (inset). (C) Uninfected E-11 cell under higher magnification showing normal mitochondria (arrow) and rER (arrowheads). (D) TiLV-infected E-11 cells under higher magnification showing normal mitochondria (arrow).

mitochondrial membrane potential as indicated by red-to-green fluorescence (R/G) ratio in TiLV-infected cells (Figs. 4C & 4D). At 0 dpi, there was no significant difference in the R/G ratio ($1.22 \pm 0.06$) in TiLV-infected cells compared to uninfected cells ($1.50 \pm 0.05$). However, at 1 dpi, the R/G ratio decreased to $1.03 \pm 0.06$ and remained at a similar level at 3 dpi ($1.012 \pm 0.04$) and 6 dpi ($1.00 \pm 0.11$). Additionally, the MitoTracker™ Red staining revealed loss of mitochondrial mass following TiLV infection at 1 dpi (Fig. 5). Comparison of the MTT assay, ATP measurement and JC-1 staining between control and TiLV-infected cells revealed statistical significance at 1, 3, and 6 dpi ($p < 0.05$). These results indicated significant damage to the mitochondria and reduction in cellular viability in E-11 cells following TiLV infection.

**Uninfected cell**          **Infected cell**

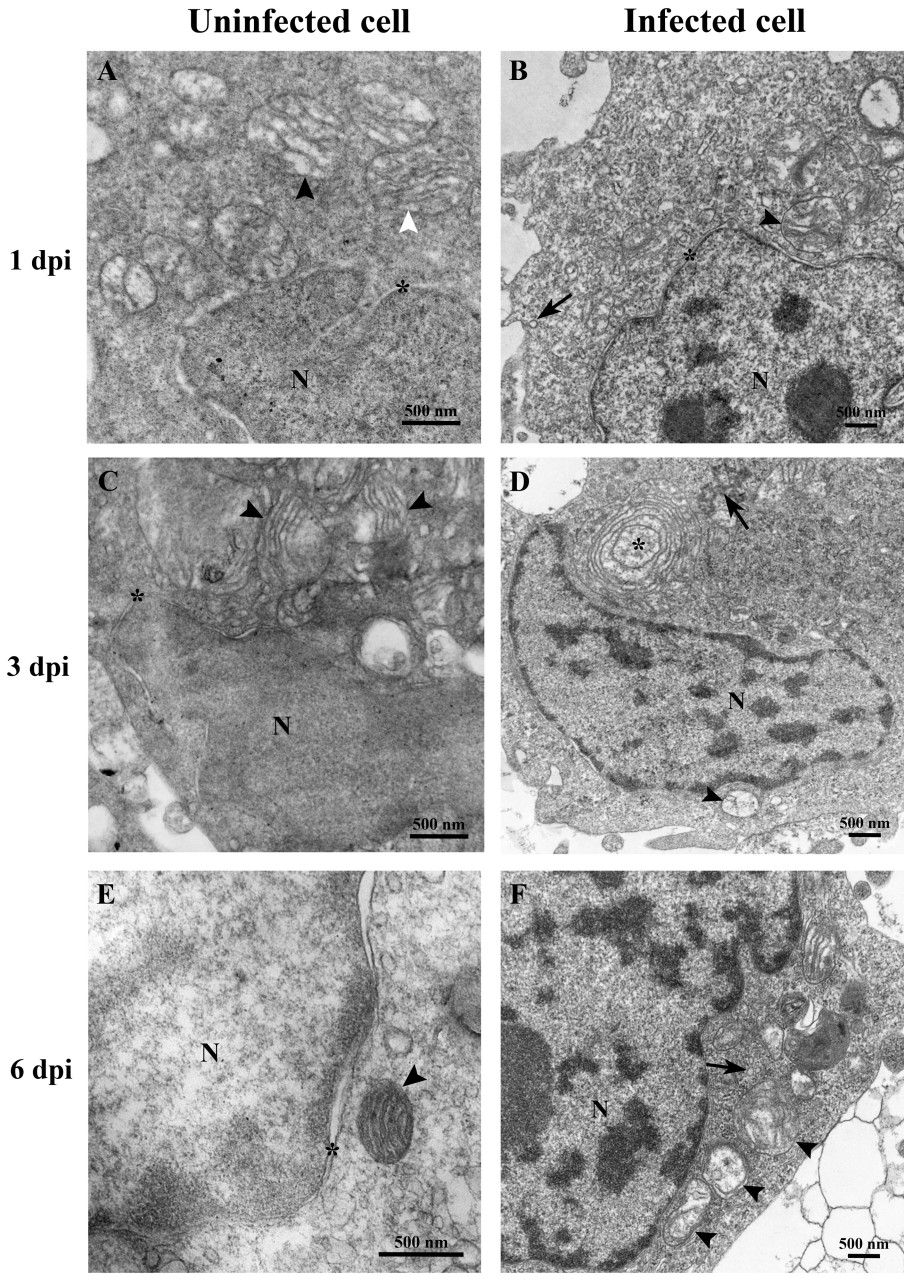

**Figure 3  Time course analysis of ultrastructural changes of TiLV-infected E-11 cells.** (A) Uninfected cell at 1 dpi presenting a normal mitochondrion with intact cristae (white arrow) and mitochondria with partial loss of cristae (black arrows). (B) TiLV-infected cell at 1 dpi, moderate mitochondrial damage (arrowhead) with the presence of a TiLV particle close to the plasma membrane (arrow). Nuclear membrane is still intact (asterisk). (C) Uninfected cell at 3 dpi displaying normal mitochondria and some mitochondria with partial loss of cristae (arrowhead), N = nucleus. (D) TiLV-infected cell at 3 dpi, a mitochondrion without cristae (arrowhead), and abundance of TiLV particles (arrow) close to lamellar bodies (asterisk) can be seen. (E) Uninfected cell with normal mitochondrion (arrowhead) near the intact nuclear membrane (asterisk), N = nucleus. (F) TiLV-infected cell at 6 dpi presenting a group of extensive degenerated mitochondria (arrowheads) surrounding TiLV particles (arrow), N = nucleus.

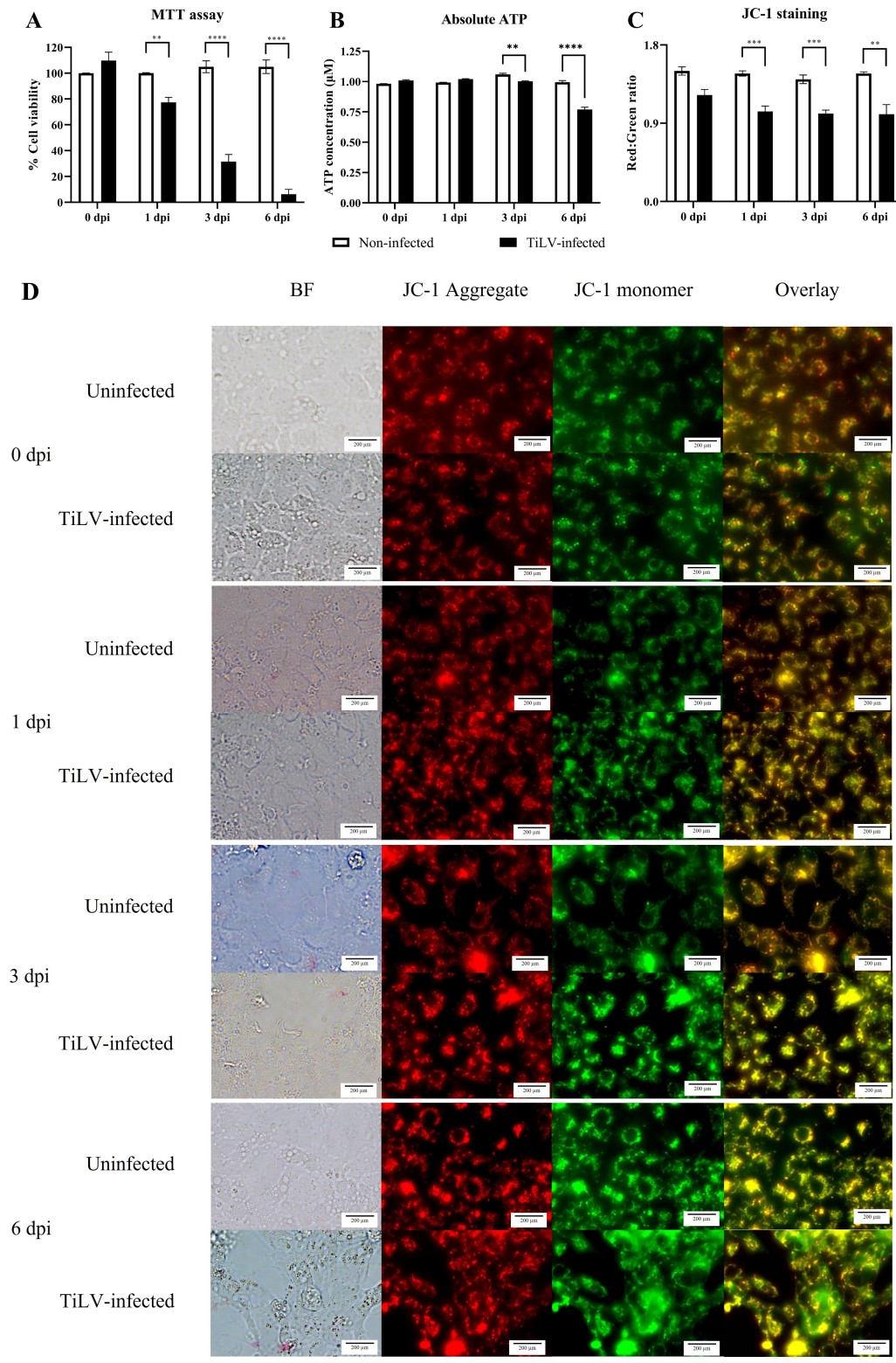

**Figure 4 Determination of mitochondrial structural and functional damage upon TiLV-infected E-11 cells at 0, 1, 3, and 6 dpi.** (A) Survival of E-11 cells after TiLV infection assessed by MTT assay. (continued on next page…)

**Figure 4 (...continued)**
(B) ATP concentration measured using CellTiter-Glo® assay (C) Mitochondrial damage in TiLV-infected cells was analyzed based on the red-to-green ratio of JC-1-stained E-11 cells. Data were quantified from three separate fields of overlay picture and shown as average values. All data were represented as the mean ± standard error of mean (S.E.M.) from three independent experiments. (D) Bright field (BF), red cells (JC-1 aggregate), green cells (JC-1 monomer), and overlay pictures of uninfected and TiLV-infected E-11 cells. Statistical significance between uninfected and TiLV-infected cells is denoted by *$p < 0.05$, **$p < 0.01$, and ***$p < 0.001$.

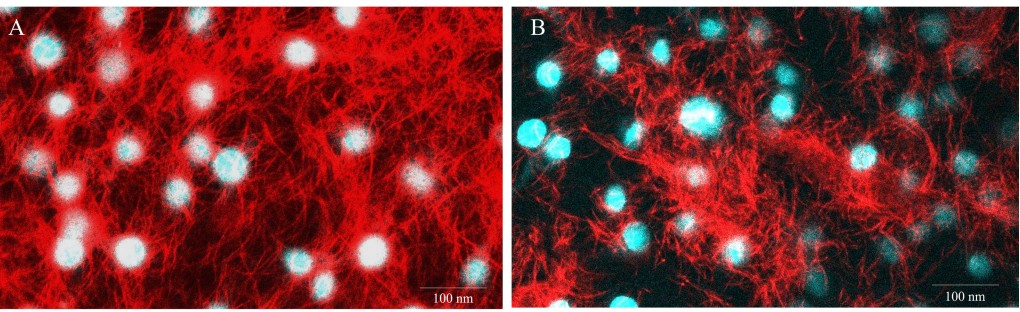

**Figure 5  Representative images of E-11 at 1 dpi incubated with Mitotracker® Red staining.** (A) Uninfected cells (B) TiLV-infected cells. The nuclei were stained with DAPI.

# DISCUSSION

TiLV is a globally significant pathogen in tilapia aquaculture, causing substantial mortality and economic losses in over 18 countries (*Eyngor et al., 2014*; *He et al., 2023*; *Surachetpong, Roy & Nicholson, 2020*; *Tran et al., 2022*). While research in this area has primarily focused on epidemiology, susceptible fish species, diagnosis, and vaccine development, the underlying mechanisms by which the virus induces cell death remain poorly understood. This study provides insights into the subcellular damage of mitochondria caused by TiLV infection, which results in a decrease in MMP, mitochondrial mass, ATP production and cell viability as indicated by mitochondrial probes and cellular viability assays. These findings suggest that mitochondrial structural and functional deterioration may be a key mechanism contributing to cell death during TiLV infection.

Previous research have demonstrated the susceptibility of various cell lines to TiLV infection, including E-11 cells, a cloned cell line derived from striped snakehead (*Channa striatus*, SSN-1) cell line (*Iwamoto et al., 2000*; *Lertwanakarn et al., 2021*), and primary tilapia (*Oreochromis* spp.) cell lines isolated from the brain, heart, and liver (*Eyngor et al., 2014*; *Li et al., 2022a*; *Li et al., 2022b*; *Yadav et al., 2021*). Furthermore, TiLV has been shown to infect primary cells from Mozambique tilapia (*O. mossambicus*) (*Kembou Tsofack et al., 2017*; *Nanthini et al., 2019*) as well as other fish cells (*Li et al., 2022a*). Despite this knowledge, the cellular damage mechanisms and physiological changes in fish cells during TiLV infection have not been extensively studied.

In this study, TEM was employed to investigate the ultrastructural changes of E-11 cells upon TiLV infection. Interestingly, intracellular viral particles were observed within

one hour of infection as previously reported in endothelial cells derived from the heart (*bulbus arteriosus*) tissue of tilapia (*Abu Rass et al., 2022*). The size and shape of TiLV were also consistent with previous descriptions, appearing as round to oval structures with a diameter of 50–120 nm and lacking a spike protein (*Del-Pozo et al., 2017*; *Eyngor et al., 2014*; *Tattiyapong, Dachavichitlead & Surachetpong, 2017*; *Yadav et al., 2021*). Similarly, TEM studies revealed the presence of TiLV particles in the liver of infected fish, in both laboratory and natural settings, although the specific mechanism of cell entry remains undetermined (*Del-Pozo et al., 2017*; *Tattiyapong, Dachavichitlead & Surachetpong, 2017*). Recent reports demonstrated that TiLV enters tilapia cells *via* a cholesterol-dependent, dynamin-mediated endocytosis mechanism (*Abu Rass et al., 2022*) and MAPK-dependent signaling pathway (*Lertwanakarn et al., 2023*). In our study, we observed notable changes in the intracellular structure and organelles of infected E-11 cells within 1 dpi. Initial changes could be observed in mitochondria including mitochondrial distortion, swelling, and loss of cristae. With further progression of the infection, the appearance of lamella bodies, as along with rapid organelle damages and cell death was found between 3 to 6 dpi. Our results are consistent with previous studies by *Del-Pozo et al. (2017)* and *Ferguson et al. (2014)*, which reported cellular and organelle damage in the hepatocytes of naturally infected fish, including an enlarged Golgi apparatus and swollen mitochondria with loss of cristae.

From our perspective, further research using other cell lines would be beneficial to replicate the findings of this study, which demonstrated that TiLV infection leads to significant damage to mitochondria, loss of its function and a decline in cellular viability. Mitochondria, as a crucial component of cellular energy production, have been found to be targeted by various viruses in both fish and mammals (*Chen et al., 2022*; *Elesela & Lukacs, 2021*; *Singh et al., 2020*; *Wang et al., 2020*). For instance, the SARS-CoV-2 virus, responsible for COVID-19, targets mitochondria and induces depolarization of the mitochondrial membrane potential, leading to the release of reactive oxygen species and greater virulence (*Shang et al., 2022*). Likewise, the hepatitis E virus targets gerbil brain tissue and causes mitochondrial damage, resulting in the disappearance of cristae and matrix (*Tian et al., 2019*). In fish, the Infectious Spleen and Kidney Necrosis Virus (ISKNV) disrupts the MMP by promoting the generation of pro-apoptotic Bax and Bak proteins and inhibiting anti-apoptotic Bcl-2 protein, leading to cell apoptosis and necrosis (*Chen et al., 2022*).

Our study provides the first evidence on the role of mitochondrial damage in the pathogenesis of cellular death during TiLV infection. The disruption of mitochondrial function was demonstrated through various assays, including ATP measurement, and the application of mitochondrial probes such as JC-1 and MitoTracker™ staining. Indeed, JC-1 and MitoTracker™ red are widely accepted and reliable methods for assessing changes in MMP and mitochondrial mass. Specifically, it was found that alterations in MMP occurred within 1 day after TiLV infection, which coincided with a reduction in the number of viable cells within the same timeframe. Additionally, a decrease in mitochondrial mass was evident in the TiLV-infected cells within 1 day, and intracellular ATP levels decreased significantly within 3 days, followed by a substantial increase in cell death. This finding is consistent with a previous report showing that the fish herpesvirus protein (CaHV-138L)

binds to the mitochondrial FoF1-ATPase and disrupt its function (*Zhao, Zeng & Zhang, 2020*). Notably, our preliminary inquiries conducted through the MitoFates database (http://mitf.cbrc.jp/MitoFates) identify the potential involvement of a hypothetical protein located within segment 6 of TiLV in mitochondrial interactions. In influenza A virus, it has been demonstrated that PB1-F2 protein promotes apoptosis in infected cells by interacting to two mitochondrial proteins, adenine nucleotide translocator 3 (ANT3) and voltage-dependent anion channel 1 (VDAC1). This interaction leads to the compromise of mitochondrial integrity and the subsequent cytochrome C release (*Zamarin et al., 2005*). Nevertheless, the mechanisms underlying mitochondrial damage, along with the specific proteins associated with TiLV and mitochondria require further in-depth investigation. Collectively, these findings highlight the crucial role of mitochondria during viral infections and emphasize the potential role of viruses to target and impair the cellular organelle.

Our novel findings align with earlier studies that have shown that mitochondria are frequently targeted by fish viruses and play a vital role in the pathogenesis and cell death process (*Chen et al., 2022*). While this study did not provide conclusive evidence that TiLV specifically invades or attaches to mitochondrial proteins, TEM revealed that viral particles were located close to the organelle and caused substantial damage during infection. Further research is necessary to fully comprehend the mechanisms by which TiLV damages mitochondria such as oxygen consumption rate, and investigation of specific signaling pathways, genes, and proteins involved in the process.

## CONCLUSIONS

In summary, our study provides evidence of the virulence and pathogenesis of TiLV, through the detection of viral particles in infected cells, the damage of mitochondria, reduction in ATP production, mitochondria mass, and cell death. Understanding the connections between mitochondrial damage and physiological disturbance in tilapia hosts, while considering other environmental factors that contribute to virus transmission, such as water quality, virus shedding, and genetic variation between viruses and hosts, is crucial for comprehending the impact of virus-host interactions on disease transmission and the fitness of TiLV in tilapia. Further research on the pathogenic mechanisms of TiLV in fish *in vivo* will expand our understanding of the virus-fish interaction.

### Funding
This project is funded by the National Research Council of Thailand (NRCT): NRCT5-RSA63002-04, Thailand. This project received financial support from the Faculty of Veterinary Medicine, Kasetsart University. The funders had no role in study design, data collection and analysis, decision to publish, or preparation of the manuscript.

### Grant Disclosures
The following grant information was disclosed by the authors:

National Research Council of Thailand (NRCT):  NRCT5-RSA63002-04.
Faculty of Veterinary Medicine, Kasetsart University.

## Competing Interests

The authors declare there are no competing interests.

## Author Contributions

- Promporn Raksaseri conceived and designed the experiments, performed the experiments, analyzed the data, prepared figures and/or tables, authored or reviewed drafts of the article, and approved the final draft.
- Tuchakorn Lertwanakarn conceived and designed the experiments, analyzed the data, prepared figures and/or tables, authored or reviewed drafts of the article, and approved the final draft.
- Puntanat Tattiyapong performed the experiments, analyzed the data, prepared figures and/or tables, authored or reviewed drafts of the article, and approved the final draft.
- Anusak Kijtawornrat analyzed the data, prepared figures and/or tables, authored or reviewed drafts of the article, provide reagents/tools, and approved the final draft.
- Wuthichai Klomkleaw conceived and designed the experiments, analyzed the data, prepared figures and/or tables, authored or reviewed drafts of the article, and approved the final draft.
- Win Surachetpong conceived and designed the experiments, performed the experiments, analyzed the data, prepared figures and/or tables, authored or reviewed drafts of the article, and approved the final draft.

## Ethics

The following information was supplied relating to ethical approvals (i.e., approving body and any reference numbers):

The protocol for handling the virus was approved by the Institutional Biosafety Committee (IBC), Faculty of Veterinary Medicine, Kasetsart University.

## Data Availability

The raw data are available in the Supplementary File.

## Supplemental Information

Supplemental information for this article can be found online at http://dx.doi.org/10.7717/peerj.16190#supplemental-information.

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
