# Peer review of "Tilapia lake virus causes mitochondrial damage: a proposed mechanism that leads to extensive death in fish cells"

_PeerJ, doi:10.7717/peerj.16190_

## Round 0.1 · original submission · Major Revisions

The subject of the paper is interesting but presents several major concerns. The first is that ultrastructural modifications in the mitochondria of infected cells are described, but no image analysis study has been performed on the images, which would allow us to really assess the differences between the two groups of cells (control and infected), so this part of the work is not complete. In addition, it would be possible to measure mitochondrial activity in real-time, which would make it possible to see the damage in the function of this organelle. Without these analyses, it is not possible to say the true effect of viral infection on mitochondria; it can only be affirmed the cellular damage at the ultrastructural level. Therefore, the work should be thoroughly revised to clarify these concepts and stick to the results obtained.

·

Basic reporting

The authors have provided sufficient background and references to research work. The article is really well structured with an overall good flow of ideas and a very clear and simple language understandable to most readers in the field.
However, the English language could need a bit of improvement in few places such as lines:
- 72 (“… that affects…” could be replaced by “currently affecting” for example)
- 75-76 (“Studies have shown that these are the same virus and can cause disease in tilapia” could be improved by writing “Subsequent studies have shown that SHT and TiLV share 98 – 100% genetic sequence identity [J. Del-pozo et al. Vet Pathology 2017], and are therefore the same virus causing disease in tilapia”)
- 182 (“Following”,…), this word could be replaced by “Next”.
- 320 (“…. a previous report that showed the fish herpesvirus protein….”) could be better phrased as “… a previous report which showed that …”
Additionally, few remarks will need attention, such as:
• Line 76: it is Tilapinevirus tilapiae and not Tilapiavirus tilapiae.
• Line 78: I believe a word is missing (The virus has a spherical (…a word is missing here…) with a trilaminar capsid-like structure)
• I suggest that you improve the way you make the connection between the rationale of the study (the justification of the study) and the existing literature. The justification of the study could be added at line 100, meaning that the previous observations of mitochondria swelling led you to investigate the effects of TiLV infection on mitochondria structure and function, for instance. Unless you have other motivations justifying this study. In which case this should be mentioned as well.
• Line 161: it should be “the pellets were rinsed with dH2O for 10 min three times” and not “The pellets were rinsed with dH2O 10 min for three times” (note the position of the word “for”)
• Line 182, I don’t think the word “Following” is the right word to use here. Maybe the word “Next” will fit better.
• Line 306: “the SARS-COV-2” not “the SAR-COV-2”

Experimental design

The here presented manuscript reports on a biological process (mitochondrial damage) occurring during viral infection with tilapia lake virus (TiLV). It is therefore within the Biological Sciences scope of PeerJ life and environment journal. In addition, the experimental design section is well organized and well-detailed. However, few aspects will require some attention:
• Under the section Materials and Methods, sub-section Propagation of TiLV, Line 130. I think a little bit should be said about the infection of the cells during virus propagation. How the cells were infected with the virus during virus propagation is completely missing. So before monitoring the progression of the infection until appearance of the CPE, I think the cells need to be infected. And this could be mentioned.

• In the sub-section Cell viability assay, you state the TCID50 (line 181) for infection, but in the sub-section Detection of red-to-green ratio in JC-1-stained E-11 cells (line 194) , you state the MOI for infection. For a bit more consistency, maybe you should choose only one of these parameters and use it consistently throughout the manuscript. And if you decide to use them both, please ensure that you also report why in one experiment you use TCID50 and in another you use MOI.

Validity of the findings

In the current manuscript, all the underlying data provided are statistically sound, with the right number of replicates to ensure statistical significance and reproducibility. The conclusions are well-stated and limited to the supporting results. I however I have few concerns.
• In Figure 2C, the image is taken at 0 dpi. But which time point post virus inoculation is that? Is it still 1hr post-TiLV inoculation? Please indicate the time point post-TiLV inoculation as you have done in Figure 2B.

• In Figure 3B (3 dpi), line 358, the abundance of TiLV particles is supposed to be indicated by an arrow. But in this image there is no arrow. Please include the arrow to make it easy for the reader to visualise the particles you are referring to, because as the image is right now, it is a bit difficult to see these particles.

• Figure 3D: extensive degeneration of mitochondria (asterisk) “close” to TiLV particles not “closes” to TiLV particles.

• About the MTT assay
The MTT assay has been found to present several limitations (Ghasemi M et al., Int. Jour. of molecular Sciences, 2021). In fact Surin AM et al. (Biochemistry (Mosc), 2017) have shown that the MTT assay, although widely used as a common tool to measure cell proliferation/viability, drug cytotoxicity, and mitochondrial/metabolic activity of cells, can disrupt the functional activity of mitochondria in cultured neurons.
So can you please explain why you used this method? A better alternative could have been an ATP assay, which measures ATP as a marker of viable cells.

Additional comments

Although your results on the JC-1 assay are compelling, explaining why cell viability decreases over time to almost zero while JC-1 staining, although dropping at the beginning of the infection, still remain constant throughout the infection, would have significantly improved the understanding of the here presented results.
The JC-1 assay remains a great tool for assessing the percentage of mitochondrial depolarization occurring in pathological and disease conditions. However, I believe we are missing a positive control to this experiment. An example of positive control could be the carbonyl cyanide m-chlorophenyl hydrazone (CCCP), which is a chemical inhibitor of oxidative phosphorylation, affecting the protein synthesis reactions in seedling mitochondria and causing the gradual destruction of living cells and death of the organism. This control is even more needed when considering that the levels of JC-1 staining appear to remain constant starting from 1 dpi onwards, although there should be more and more less viable cells as the infection progresses (as even supported by the results of your MTT assay).

Moreover, the paper could have really benefitted from images of the JC-1 staining at higher magnification, to enable a better visualization of the colocalization of the green and red by the reader.

From my understanding of the introductory section, this study was undertaken to contribute to the limited understanding of the biology and host cell alteration caused by TiLV infection. More specifically to provide insights into the pathogenesis of this virus and improve the understanding of the mechanism by which TiLV causes cell damage, changes in cellular functions and cell death (for a better development of strategies to prevent and manage TiLV infections in fish populations).
As such, the addition of a mitochondrial stress assay (to measure key parameters of mitochondrial function such as oxygen consumption rate) for instance, to the overall work done in this study, could have given more information on how the overall mitochondrial metabolism is affected by the infection and would have thus reveal more information on the biology of the virus and the effect of the infection on the cellular metabolism.

·

Basic reporting

The manuscript is well written, clear and unambiguous.

Experimental design

Research question well defined, relevant & meaningful. Novelty of the study should be stressed.

Validity of the findings

1) The novelty of the study should be stressed in the introduction or discussion.
2) The conclusion section of the abstract is more like a statement of the significance of the study.
3) Quality of Figure 2A and B is very low, it is hard to identify typical mitochondria from these photos.

Additional comments

1) Figure 3B, there is not an arrow indicating the TiLV particles.
2) Line 80, Cichlid should be cichlid.
3) Line 120, MTT assay is not dedicated to analyze mitochondria structure and function
4) Line 277, a cloned cell line of striped snakehead

---

## Round 0.2 · Minor Revisions

Thank you for completing and modifying the initial manuscript. Although it has been substantially improved, there are still some bugs that need to be improved and have been suggested by the reviewer.

I hope you will take all these suggestions into account in the next revised manuscript.

Best regards,

·

Basic reporting

The manuscript is now better written, and the authors have now made a better connection between the rationale of the study (the justification of the study) and the existing literature. It is an overall well-structured paper.
Few mistakes still subsist though, but these can be easily addressed.
These include:
• Line 45: that currently affects (not affecting)

• Line 49: a properly written citation of Del-pozo et al. should be written (without the journal name)

• Line 76: “….dynamic studies have never…” (not had never)

• Line 214 - 215: The language in these lines is a bit confusing (… the data were test normal distribution using …).
I believe a word is missing here or the sentence should be phrased differently. Maybe the authors wanted to say: “The data were tested for normal distribution using…” or “the normal distribution of the data was tested using…”. Please clarify this sentence.
• Line 223 and 235: Throughout the text you have been using “h” for hour. But in these 2 lines you are no longer using the abbreviation “h” but rather the full form “hours”. It will be better to be consistent I think.

• Line 281 – 282: You already said, “subcellular damage of mitochondria…”, I don’t think you should add again “specifically targeting the mitochondria”. You are already saying that the damage is on the mitochondria.
Alternatively, you could say “…subcellular damage of mitochondria caused by TiLV infection. The specific targeting of mitochondria by TiLV results in a decrease in MMP…”. Note the presence of the full stop, meaning that these now become two separate sentences.

• Line 340 – 341: “…located close to …” (remove the word “in”).

• Line 342: “…to fully comprehend the mechanism…” either “through which” or “by which”, but not both. So choose either “through” or “by”.

Experimental design

The experimental design section is well organized and now sufficiently well-detailed. The right number of replicates have been carried out, and the units used for virus infection are now consistent throughout the paper.

Validity of the findings

In the current version of the paper, the underlying data provided are statistically sound, and as previously mentioned, the right number of replicates have been done. The conclusions are safely well-stated meaning that they are limited to the supporting results.
The additional ATP assay as well as the MitotrackerTM Red staining indeed provide further evidence of an impairment of the mitochondria function (and integrity), thus further consolidating the initial results presented by the authors.

Additional comments

• Line 296: The cell line used in this study by Abu Rass et al. (2022) is TmB. These are endothelial cells deriving from the heart (bulbus arteriosus) tissue of tilapia and not from the brain (as you are saying). So TmB are not brain cells. This information should be corrected.

• Line 241 (and Figure 3 as well). What can explain the partial loss of cristae observed in the uninfected cells at 1 dpi? Can you please give a little bit of justification as to why this happens?
Just as a general comment (or question), have you tried to search for possible mitochondrial targeting sequences in the hypothetical protein sequences of TiLV? Including in the alternative reading frames deriving from TiLV hypothetical proteins.
This can easily be done using several servers, for example PredictProtein server, MitoFates, TPpred2 server or iPSORT.
The presence of a mitochondrial targeting sequence in one (or several) protein sequence(s) of TiLV could also suggest the mitochondrial localization of that viral protein and could thus potentially explain the mechanisms by which TiLV infection causes mitochondrial impairment, which as you say plays a significant role in the biology and pathogenicity of the virus.
It has for instance been demonstrated that PB1-F2 of influenza A virus, is a mitochondrial accessory protein of PB1. And by localizing within the mitochondria during the infection, PB1-F2 promotes apoptosis in infected cells by disrupting mitochondrial integrity and by releasing cytochrome C (Zamarin et al. 2005, Influenza virus PB1-F2 protein induces cell death through mitochondrial ANT3 and VDAC1. PLoS Pathog 1:e4. http://dx.doi.org/10.1371/journal.ppat.0010004) . So maybe a similar pathway occurs during TiLV infection? Maybe, maybe not.

---

## Round 0.3 · accepted · Accept

Thank you for submitting your work to this journal.

With kind regards,

·

Basic reporting

The manuscript is now better written, and the few mistakes pointed out have now been addressed. It is therefore better detailed and well-structured

Experimental design

The experimental design section is also well organized and now sufficiently well-detailed. The right number of replicates have been carried out, and the units used for virus infection are now consistent throughout the paper.

Validity of the findings

In the current version of the paper, the underlying data provided are statistically sound, and as previously mentioned, the right number of replicates have been done. The conclusions are safely well-stated meaning that they are limited to the supporting results.
The additional ATP assay as well as the MitotrackerTM Red staining indeed provide further evidence of an impairment of the mitochondria function (and integrity), thus further consolidating the initial results presented by the authors.

Additional comments

I am pleased with the changes made by the authors and their willingness to correct and make this manuscript better.

·

Basic reporting

The manuscript is well written, clear and unambiguous. Professional English is used throughout.

Experimental design

The research question is well defined. The experimental design section is well organized and clearly described.

Validity of the findings

The conclusions is well supported by the provided data.